# Vibrationally excited molecular hydrogen production from the water photochemistry

Yao Chang [1,6], Feng An[2,6], Zhichao Chen[1,6], Zijie Luo [1], Yarui Zhao[1], Xixi Hu [3✉], Jiayue Yang[1], Weiqing Zhang[1], Guorong Wu [1], Daiqian Xie [2], Kaijun Yuan [1,4✉] & Xueming Yang [1,5]

Vibrationally excited molecular hydrogen has been commonly observed in the dense photo-dominated regions (PDRs). It plays an important role in understanding the chemical evolution in the interstellar medium. Until recently, it was widely accepted that vibrational excitation of interstellar $H_2$ was achieved by shock wave or far-ultraviolet fluorescence pumping. Here we show a further pathway to produce vibrationally excited $H_2$ via the water photochemistry. The results indicate that the $H_2$ fragments identified in the $O(^1S) + H_2(X^1\Sigma_g^+)$ channel following vacuum ultraviolet (VUV) photodissociation of $H_2O$ in the wavelength range of $\lambda = {\sim}100\text{-}112$ nm are vibrationally excited. In particular, more than 90% of $H_2(X)$ fragments populate in a vibrational state $v = 3$ at $\lambda{\sim}112.81$ nm. The abundance of water and VUV photons in the interstellar space suggests that the contributions of these vibrationally excited $H_2$ from the water photochemistry could be significant and should be recognized in appropriate interstellar chemistry models.

[1] State Key Laboratory of Molecular Reaction Dynamics and Dalian Coherent Light Source, Dalian Institute of Chemical Physics, Chinese Academy of Sciences, 457 Zhongshan Road, Dalian 116023, China. [2] Institute of Theoretical and Computational Chemistry, Key Laboratory of Mesoscopic Chemistry, School of Chemistry and Chemical Engineering Nanjing University, Nanjing 210093, China. [3] Kuang Yaming Honors School, Institute for Brain Sciences, Nanjing University, Nanjing 210023, China. [4] University of Chinese Academy of Sciences, Beijing 100049, China. [5] Department of Chemistry, College of Science, Southern University of Science and Technology, Shenzhen 518055, China. [6] These authors contributed equally: Yao Chang, Feng An, Zhichao Chen. ✉email: xxhu@nju.edu.cn; kjyuan@dicp.ac.cn

Vibrationally excited molecular hydrogen ($H_2(v > 0)$) is a key species in the interstellar medium (ISM)[1]. The internal energy available in vibrationally excited $H_2$ can be used to overcome or diminish the activation barrier of various chemical reactions of interest for molecular astrochemistry. The enhancement in the reactivity of $H_2$ when it is in a vibrationally excited state has solved several puzzling problems, e.g., the formation of the methylidine cation ($CH^+$) in the ISM[2–4]. $CH^+$ was one of the first molecules observed in space, its ubiquity and high abundance have been a longstanding problem for more than 70 years[5]. The reaction that forms the $CH^+$ molecule is endothermic (with a barrier of ~0.37 eV). Such endothermicity could be overcome by the $C^+$ reacting with the vibrationally excited $H_2$[6,7]. Therefore, the excitation of interstellar $H_2$ is an essential prerequisite for determining the chemical composition in the ISM.

Recent astronomical observations have illustrated that conspicuous emissions from vibrationally excited $H_2$ exist in the photo-dominated regions (PDRs) and shocks[8]. The first ultraviolet detection of vibrational excited interstellar $H_2$ was performed by Federman et al. using the Hubble Space Telescope toward the star ζ Ophiuchi[9], and then detected by Jensen et al. toward HD 38087 and HD 199579[10], Gnacinski et al. towards HD 147888[11], and Racheford et al. near Herschel 36[12]. In particular, over 500 interstellar $H_2$ absorption lines from excited vibrational levels $v = 1–14$ were reported by Meyer et al. toward HD 37903, one of the hot stars located in the NGC 2023 reflection nebula[13].

It was widely accepted that two major sources were responsible for the excitation of $H_2$ in the ISM, i.e., shock waves and far-ultraviolet (FUV) fluorescence[14,15]. For shocks, the $H_2$ molecules were collisionally excited from the ground state in shock-heated gases; whereas for fluorescence they were populated by radiative decay of electronically excited states, which were pumped through the absorption of FUV photons. Though there was general agreement between the main characteristics of models and observations for $H_2$ excitation, some discrepancies are yet to be unraveled. For instance, the observations showed that the $v = 4$ lines are significantly stronger than the predictions[16]. A possible explanation for an excess in $v = 4$ was that $H_2$ formation is occurring directly into this level, which inspired the scientists to look for processes other than collisions and fluorescence excitation in PDRs. In this work, we demonstrated that vibrationally excited $H_2$ can be directly produced from the water photochemistry in the vacuum UV (VUV) region, via a new discovered fragmentation channel $O(^1S) + H_2$.

Water photochemistry has been of great interest to chemists since it represents a prototype system for the photodissociation of triatomic molecules, as well as an important process occurring in planetary atmospheres, interstellar clouds, and comet coma[17–20]. The electronic spectrum of $H_2O$ displays two continuous absorption bands with their maxima at $λ \sim 167$ nm and ~128 nm respectively, and stronger absorption features at $λ < 124$ nm associated with excitations to Rydberg states (Supplementary Fig. 1). Various photon-induced fragmentation processes of $H_2O$ have been studied previously[20]:

$$H_2O(\tilde{X}^1A_1) + hν \rightarrow H + OH(X^2\Pi)\,(D_{th} = 5.102\,\text{eV}) \quad (1)$$

$$\rightarrow H + OH(A^2\Sigma^+)\,(D_{th} = 9.124\,\text{eV}) \quad (2)$$

$$\rightarrow O(^3P) + H + H\,(D_{th} = 9.513\,\text{eV}) \quad (3)$$

$$\rightarrow O(^1D) + H + H\,(D_{th} = 11.480\,\text{eV}) \quad (4)$$

where the threshold energies ($D_{th}$) for these fragmentation channels are given in parentheses[21,22] (Some are on the basis of thermodynamic calculations with the data available from the thermochemical network) (https://atct.anl.gov). Photodissociation

studies at both 193 nm[23] and 157.6 nm[24–27], which excite $H_2O$ to the $\tilde{A}^1B_1$ state, reveal prompt O-H bond fission and formation of ground-state $OH(X^2\Pi)$ radical with little internal excitation. In contrast, photodissociation of $H_2O$ in its second ($\tilde{B}^1A_1 \leftarrow \tilde{X}^1A_1$) absorption band is much more complicated. This dissociation proceeds via two main pathways: one leading to the $OH(A^2\Sigma^+) + H$ products when the dissociation energy is above the threshold for this channel; the other to $OH(X^2\Pi) + H$ through two conical intersections (CIs) between the $\tilde{B}$ state and the ground state ($\tilde{X}^1A_1$) at the linear geometries HOH and OHH[28–34]. At shorter wavelengths, three body channels (3) and (4) can be reached. Recent photofragment translational spectroscopy (PTS) measurements of the H atoms from $H_2O$ photolysis in the wavelength range of $90 \leq λ \leq 110$ nm revealed the dominated production of channels (3) and (4)[35]. Of particular astrochemical significance, the PTS study showed that ~20% of $H_2O$ photo-excitation events induced by the solar photons yield O atoms via the three-body channels.

The competing $H_2$ elimination channels following VUV photoexcitation of $H_2O$ should also be thermodynamically accessible,

$$H_2O(\tilde{X}^2A_1) + hν \rightarrow O(^1D) + H_2(X^1\Sigma_g^+)\,(D_{th} = 6.983\,\text{eV}) \quad (5)$$

$$\rightarrow O(^1S) + H_2(X^1\Sigma_g^+)\,(D_{th} = 9.206\,\text{eV}) \quad (6)$$

To date, only the branching ratio for the channel (5) has been measured a long time ago by Slanger and Black[36]. Whereas the possible fragmentation channel (6) has yet to be identified in any detail. Recent development of the intense VUV free-electron laser (FEL), at the Dalian Coherent Light Source, has provided a unique tool for studying the state-of-the-art molecular fragmentation dynamics in the VUV range[37,38].

Here, we present the experimental verification of $O(^1S) + H_2(X^1\Sigma_g^+)$ fragments based on VUV-pump and VUV-probe time-sliced velocity-map imaging (TSVMI) measurements of $O(^1S)$ atoms (Fig. 1). The results indicate that all of the $H_2$ fragments identified following VUV photodissociation of $H_2O$ in the wavelength range of $λ = \sim100–112$ nm are vibrationally excited. In particular, more than 90% of $H_2(X)$ fragments populate in a single vibrational state $v = 3$ at $λ \sim 112.81$ nm, representing an alternative source of vibrationally excited $H_2$ in the ISM.

## Results and discussion

**Observation of vibrationally excited $H_2$ from $H_2O$ photolysis.** The present experimental studies were performed on the recently constructed VUV FEL-TSVMI apparatus which equips with two independently tunable VUV laser radiation sources (Fig. 1b). The VUV FEL output (Fig. 1a) was used to directly excite $H_2O$ molecules to a dissociative rovibronic state. The subsequent $O(^1S)$ fragments were then resonantly ionized at $λ = 121.7$ nm, which was generated by a table-top VUV source from the four-wave mixing scheme (see Method section for more details). A pulsed supersonic molecular beam of 3% $H_2O$/Ar was crossed by the two counter-propagating VUV beams in the interaction region of the TSVMI setup, as shown schematically in Fig. 1.

Figure 2 displays time-sliced ion images of the $O(^1S)$ photofragments recorded following photolysis of $H_2O$ at $λ = 102.67$, 105.52, 107.65, 109.12, 111.48, and 112.81 nm, respectively. These wavelengths locate in the central of the sharp features in the absorption spectrum (Supplementary Fig. 1), which are attributed to $nd \leftarrow 1b_1$ ($n \geq 3$) Rydberg transitions. The double-headed red arrows in the image figures stand for the polarization direction of photolysis lasers. At all six photolysis wavelengths, well-resolved concentric ring structures with different intensities are clearly observed in the experimental

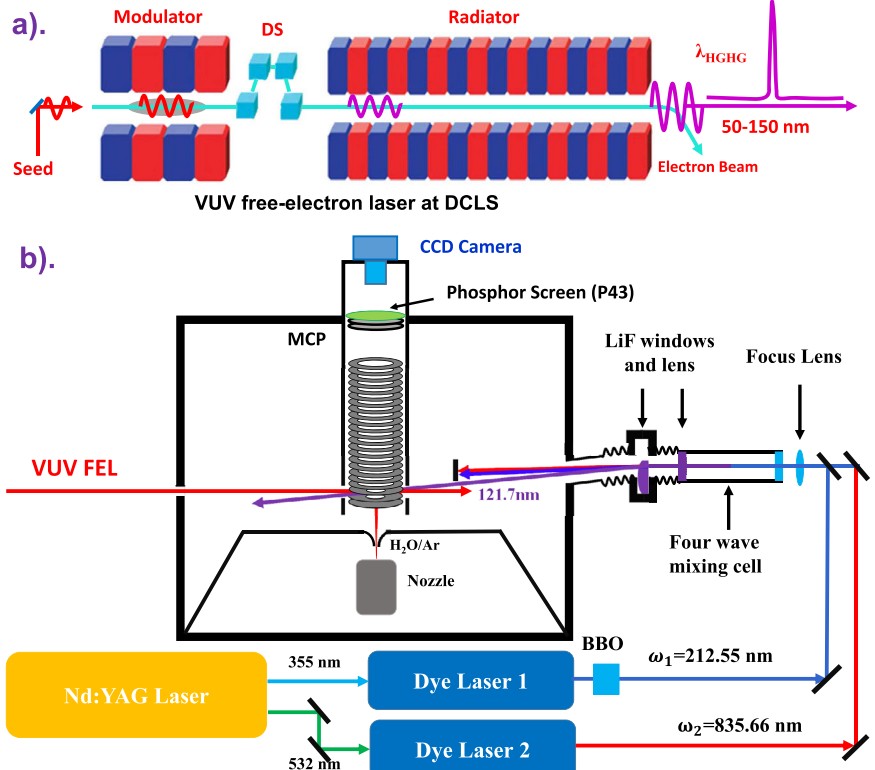

**Fig. 1 Schematic diagram of the experimental setup. a** Schematic of the VUV free-electron laser (VUV FEL) beamline at the Dalian Coherent Light Source (DCLS). **b** The arrangement of VUV-pump and VUV-probe time-sliced velocity-map imaging (TSVMI) system for the water photochemistry. Tunable VUV pumping source ($\lambda_1 \sim 100$–112 nm) comes from VUV FEL beamline, and the fixed VUV probing light ($\lambda_2 = 121.7$ nm) is generated by the two-photon resonance-enhanced four-wave mixing ($\omega(\lambda_2) = 2\omega_1 - \omega_2$) scheme with dispersing the fundamental radiations ($\omega_1$ and $\omega_2$) from the direction of the VUV radiation. DS: dispersion section; MCP: micro-channel plate; BBO: beta barium borate crystal; HGHG: high gain harmonic generation.

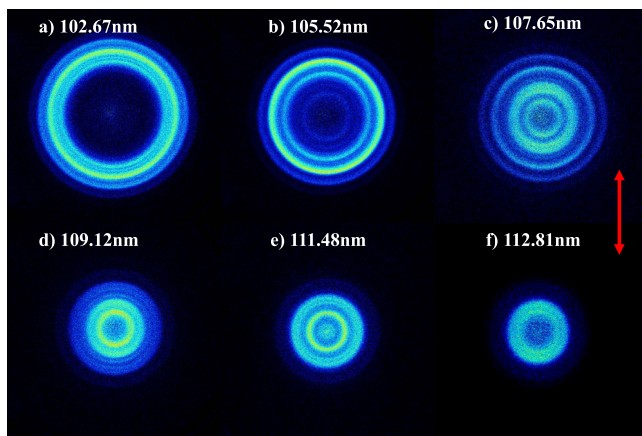

**Fig. 2 Wavelength dependent time-sliced velocity map images from $H_2O$ photolysis.** Time-sliced images of the $O(^1S)$ photoproducts from photodissociation of $H_2O$ at **a** 102.67, **b** 105.52, **c** 107.65, **d** 109.12, **e** 111.48, and **f** 112.81 nm. The red double arrow indicates the polarization direction of the dissociation laser. The ring features correspond to the rovibrational states of the coincident $H_2(X, v)$ products.

images, which can be directly attributed to different vibrational states of $H_2(X)$ coproducts. It is noted that these structures cannot arise from the other $O(^1S)$ elimination process, i.e., the $O(^1S) + H + H$ channel, because of the insufficient photolysis photon energy ($D_{th} \geq 13.703 \text{eV}$[22]). In addition, an off-axis biconvex LiF lens was used to disperse the 212.55 nm and 835.66 nm lights from the photodissociation/photoionization region, ensuring no secondary dissociation (e.g., the primary

OH(X/A) fragment subsequently absorbs another UV photons and dissociates) takes place. The two-photon excitation process is also not possible by applying the extremely low intensity of the VUV light.

The velocity distributions of the $O(^1S)$ fragments were determined from the radii of the resolved ring structures in the VMI images, from which the total kinetic energy release distributions ($E_T$) were derived and shown in Fig. 3. In this photodissociation experiments, the available energy (the photo-excitation energy subtracts the threshold energy ($h\nu - D_{th}$)) is distributed between the $O(^1S)$ and $H_2(X)$ product kinetic energy and the internal energy ($E_{int}$). Then the internal energy distributions of $H_2(X)$ fragments can be deduced from the $E_T$ spectra by using the law of energy conservation,

$$h\nu(\lambda_{\text{VUVFEL}}) - D_{th} = E_{int}[O(^1S)] + E_{int}[H_2(X)] + E_T[O(^1S) + H_2(X)]$$

$$(7)$$

The energy combs representing the vibrational quantum numbers of $H_2(X)$ fragments were labeled in Fig. 3. The structures provide $H_2$ vibrational distributions with vibrational and partially rotational resolution. Best-fit simulations of these spectra return $H_2(X, v)$ population distributions (Fig. 4 and Table 1). It is notable that all of the $H_2$ fragments formed at the six photolysis wavelengths are vibrationally excited with cold rotational excitation ($J < 9$, Supplementary Fig. 2), and the inverted vibrational state population distributions mainly span vibrational levels of $1 \leq v \leq 5$, peaking at $v = 4$ at 109.12 nm and 107.65 nm, and $v = 3$ at other wavelengths. The most striking finding is that more than 90% of $H_2(X)$ fragments populate to a single excited vibrational state $v = 3$ at 111.48 and 112.81 nm,

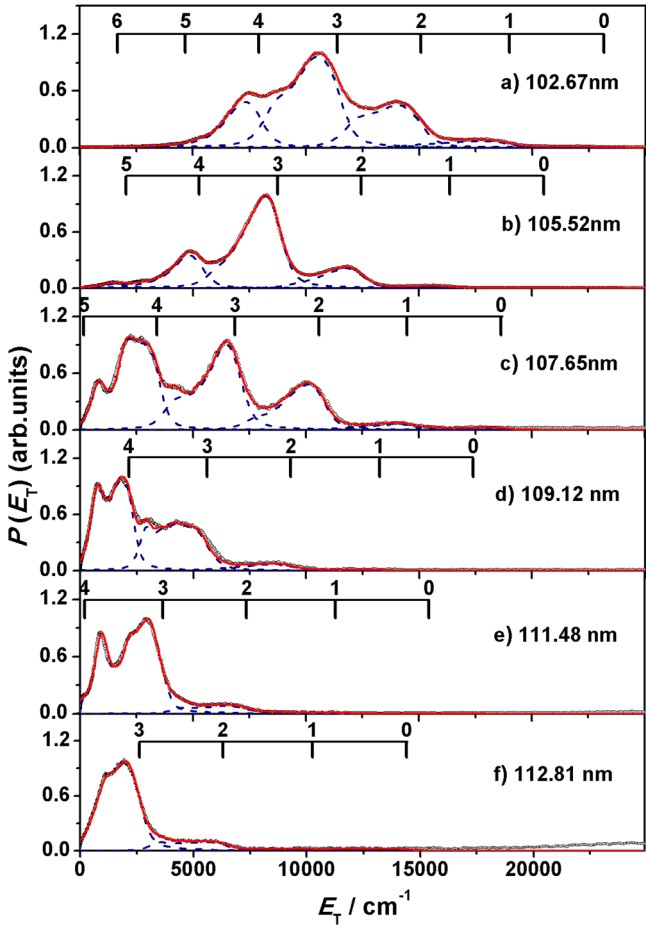

**Fig. 3 Wavelength-dependent product kinetic energy distributions from $H_2O$ photolysis.** The product total kinetic energy distribution ($E_T$) spectra derived from the images shown in Fig. 2, in red, along with the best-fit simulation of the spectra, in navy. The superposed combs indicate the $E_T$ values associated with formation of the various vibrational states of $H_2(X)$. The raw data are provided as a Source Data file.

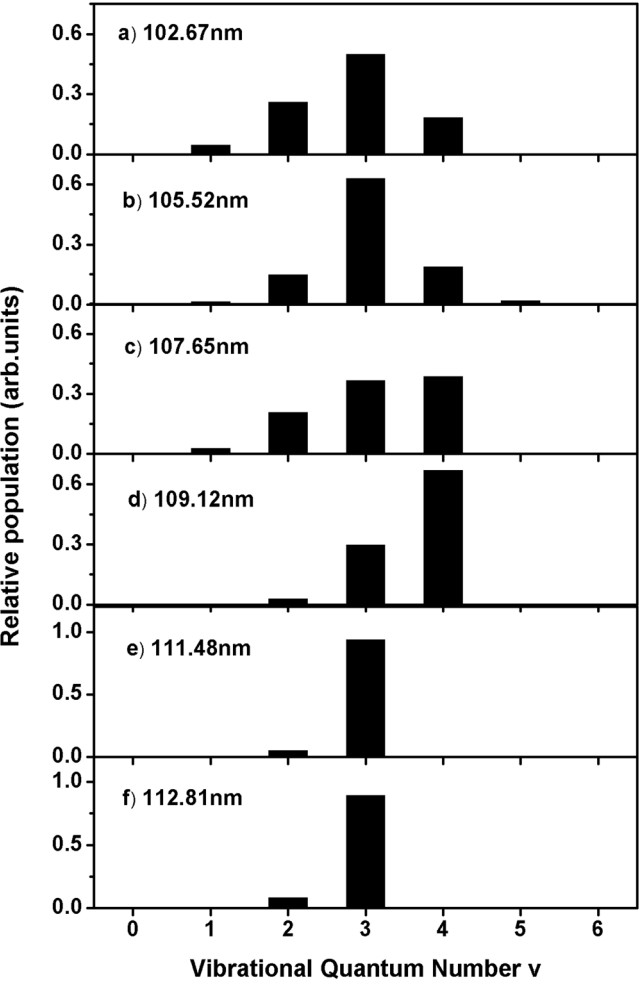

**Fig. 4 Vibrational state distributions of $H_2$ from $H_2O$ photolysis.** Relative populations of different vibrational states of the $H_2(X)$ products in the $O(^1S) + H_2$ channel from the $O(^1S)$ detection at six photolysis wavelengths. The raw data are provided as a Source Data file.

suggesting peculiar dissociation dynamics. This phenomenon has a little similarity with the previously reported ClO vibrational distribution from OClO photodissociation[39] and the OD rotational distribution from HOD photodissociation[40]. Further Analysis (Supplementary note 1) reveals that about 80% of the available energy, or 11,450 cm$^{-1}$, is released in vibration of $H_2$ at 112.81 nm photolysis, the rest of the energy is partitioned in translation (~18%) and rotation (~2%). As the photoexcitation energy increases, however, almost the same energy ~11,100 cm$^{-1}$ is released in vibration, e.g., at 102.67 nm, implying the dissociation undergoes the same transition state. The excess of the available energy tends to depositing in translation (~47% at 102.67 nm).

To further reveal this dissociation process, photodissociation of $D_2O$ has also been investigated using the same strategy. Supplementary Fig. S3 shows time-sliced velocity map images of the $O(^1S)$ fragments following photoexcitation of $D_2O$ at $\lambda = 98.80$, 100.36, 102.65, 105.21, 108.29, 111.29, and 112.71 nm, respectively. Image analysis yields the corresponding $P(E_T)$ distributions (Supplementary Fig. S4), and best-fit simulations of these spectra return $D_2(X, v)$ population distributions (Supplementary Fig. S5 and Table 1). In similarity with the $H_2$ fragments from $H_2O$ photolysis, the $D_2$ fragment vibrational state population distributions mainly possess vibrational levels of $2 \leq v \leq 7$, peaking at $v = 4$ at 112.71 nm, and $v = 5$ at other

wavelengths. The single $v$ propensity has also been observed, with 95% of $D_2(X)$ fragments populating to a single excited vibrational state $v = 4$ at 112.71 nm, which means ~11,108 cm$^{-1}$ of the available energy is released in vibration of $D_2$.

The $O(^1S)$ ion image measurements reveal not only the $E_T$ distribution of the photo-product channels, but also the angular distribution of the photo-fragments, which is characterized by the anisotropy $\beta$ parameter. The $O(^1S)$ ion images display near isotropic distributions with average parameters $\beta = \sim 0–0.2$ at these photolysis wavelengths, indicating that the dissociation process should be quite slow. All of these observations could be accounted for qualitatively by a transition state structure in which both hydrogen atoms are on the same side of the O atom, or the two hydrogen atoms are close enough ensuring two O-H bonds simultaneously break and two H atoms recombine to form $H_2$, as illustrated in $CO_2$ dissociation[41].

The calculated energy diagram for the vertical excitation of $H_2O$ and the subsequent pathways for the $O + H_2$ dissociation channels obtained from the potential energy surfaces (PESs) are shown in Fig. 5, and in Supplementary note 2. According to the previous studies[35], photoexcitation at $\lambda \sim 100–112$ nm which populates the $nd$ Rydberg states can undergo efficient non-adiabatic coupling to the $\tilde{E}'$ and $\tilde{D}$ electronic states. There are two pathways leading to the $O(^1S) + H_2 (X^1\Sigma_g^+)$ fragments. One is an

**Table 1 The relative vibrational state populations of H₂ (D₂) photofragments for the O(¹S) + H₂ (D₂) channel following H₂O (D₂O) photolysis in the wavelengths range of 100–112 nm. The sum populations of all vibrational states have been normalized to unity.**

| H₂/D₂(v) | H₂O (λ/nm) | | | | | | D₂O (λ/nm) | | | | | | |
|---|---|---|---|---|---|---|---|---|---|---|---|---|---|
| | 102.67 | 105.52 | 107.65 | 109.12 | 111.48 | 112.81 | 98.80 | 100.36 | 102.65 | 105.52 | 108.51 | 111.29 | 112.71 |
| 0 | 0 | 0 | 0 | 0 | 0 | 0 | 0 | 0 | 0 | 0 | 0 | 0 | 0 |
| 1 | 0.05 | 0.01 | 0.03 | 0 | 0 | 0 | 0 | 0 | 0 | 0 | 0 | 0 | 0 |
| 2 | 0.26 | 0.15 | 0.21 | 0.03 | 0.06 | 0.09 | 0.05 | 0.05 | 0.02 | 0 | 0 | 0 | 0.01 |
| 3 | 0.50 | 0.63 | 0.37 | 0.30 | 0.94 | 0.91 | 0.14 | 0.15 | 0.07 | 0.04 | 0.02 | 0.02 | 0.04 |
| 4 | 0.19 | 0.19 | 0.39 | 0.67 | | | 0.24 | 0.27 | 0.20 | 0.29 | 0.12 | 0.38 | 0.95 |
| 5 | 0 | 0.02 | | | | | 0.35 | 0.33 | 0.48 | 0.40 | 0.61 | 0.60 | 0 |
| 6 | 0 | | | | | | 0.19 | 0.18 | 0.22 | 0.13 | 0.25 | | |
| 7 | | | | | | | 0.03 | 0.02 | 0.01 | 0.14 | | | |
| 8 | | | | | | | 0 | 0 | 0 | 0 | | | |

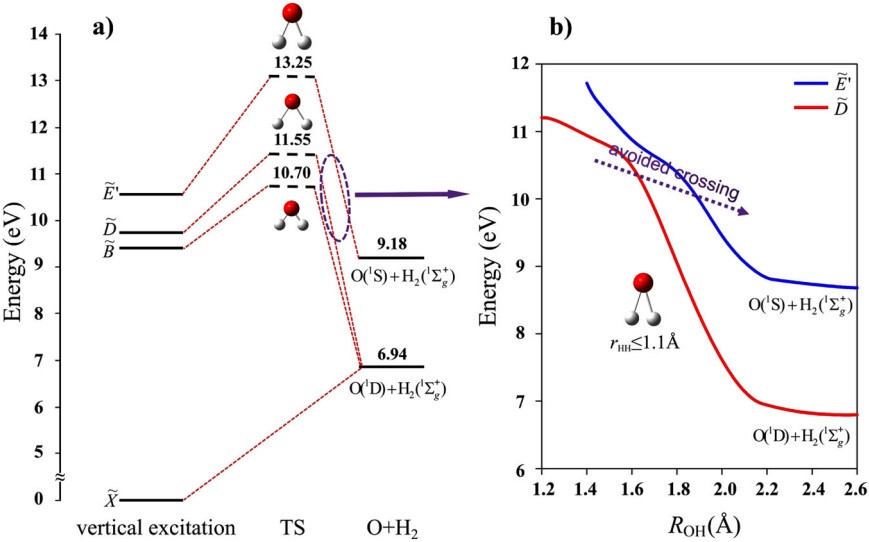

**Fig. 5 Schematic diagram of photodissociation mechanisms of H₂O. a** The correlation diagram of O + H₂ fragmentation processes from the H₂O photodissociation. The zero energy is located at the vibrational state (000) of the ground electronic state. The zero-point energy is considered. **b** The diagram of the nonadiabatic dissociation pathway from $\tilde{D}$ to $\tilde{E}'$ state through the avoid crossing on the potential energy surfaces (PESs). The purple dotted arrow indicates the region of the avoid crossing on the PESs.

**Table 2 The calculated energies and average bond lengths of vibrational states for H₂(X¹Σ_g⁺) and D₂(X¹Σ_g⁺).**

| v | H₂ | | D₂ | |
|---|---|---|---|---|
| | Energy/cm⁻¹ | $\bar{r}$/Å | Energy/cm⁻¹ | $\bar{r}$/Å |
| 0 | 0 | 0.77 | 0 | 0.76 |
| 1 | 4150.73 | 0.83 | 2987.73 | 0.81 |
| 2 | 8061.93 | 0.89 | 5852.42 | 0.85 |
| 3 | 11742.15 | 0.96 | 8594.49 | 0.89 |
| 4 | 15201.33 | 1.03 | 11219.53 | 0.94 |
| 5 | 18442.3 | 1.1 | 13735.48 | 0.98 |
| 6 | 21461.76 | 1.17 | 16147.29 | 1.03 |
| 7 | 24251.34 | 1.26 | 18453.25 | 1.08 |
| 8 | 26799.79 | 1.35 | 20649.27 | 1.14 |
| 9 | 29092.34 | 1.45 | 22730.63 | 1.19 |
| 10 | 31141.99 | 1.56 | 24694.03 | 1.25 |

adiabatic pathway via the $\tilde{E}'$ state. An extremely high barrier of 13.25 eV, however, prevents the direct dissociation from this state. Meanwhile, the PESs present a nonadiabatic pathway through an avoided crossing between the $\tilde{D}$ and $\tilde{E}'$ states, and this pathway has a much lower barrier of 11.55 eV, as shown in Fig. 5b. The avoided crossing between the two states is located where the energy difference is minimal and the wavefunctions are strongly mixed. This transition state has a geometry with the H-H bond distance shorter than 1.1 Å and the two O-H bond lengths around 1.8 Å. The calculated energies and average bond distances of vibrational states for H₂(X)/D₂(X) are shown in Table 2. When the H-H bond lengthens to ~1.1 Å, the vibrational excitation level of H₂ can be up to 5, which is inconsistent with the vibrational state population distributions of $v \leq 5$ observed in experiments. Similarly, the transition state with the D-D bond distance (~1.1 Å) leads to the vibrational excitation of D₂ fragments spanning $v \leq 7$, which is also in accord with the experimental observation. For the photoexcitation energy being lower than 11.55 eV, the water molecule can further transfer to the $\tilde{B}$ state through an avoid crossing between the $\tilde{D}$ and $\tilde{B}$ states[28], and then return to the transition state around the $\tilde{D}/\tilde{E}'$ coupling. It is noted that this central atom elimination channel has also been observed previously in the photodissociation of OCS[42], CO₂[43], and CS₂[44], where the dissociation mechanism has been illustrated as the molecular isomerization with one side atom moving to the other side via a nonadiabatic transition or a roaming process. In

comparison, the $O(^1S) + H_2$ channel mainly proceeds to the transition state with the two H atom being close, and the dissociation process involves the O-H bonds breaking and H-H bond forming. This dissociation process undergoes several internal conversions with the more scrambling that leads to near isotropic anisotropy.

**Implications for vibrationally excited $H_2$ in the ISM.** Since the rate constant enhancement when $H_2$ goes from the ground to an excited vibrational state gets larger at low temperature, the chemistry of vibrationally excited $H_2$ has been considered, not merely in the context of trying to explain the longstanding problem of $CH^+$ formation in diffuse clouds, but also in the determination of the global chemical composition in dense PDRs[1]. In addition, the vibrationally excited $H_2$ in the ground electronic state formed in rarefied interstellar environments can survive in several days (it decays radiatively with Einstein A-coefficient $A \sim 10^{-6}\,s^{-1}$)[45], making them a key species in the ISM. Vibrationally excited $H_2$ has been commonly observed in the interstellar circumstances, as in PDRs where the $H_2$ ($v > 0$) molecules are considered to be produced by FUV fluorescence, and in shocked gas where the excitation is mainly collisional. Astronomical observations, however, displayed that the $H_2$ molecules have significant populations in high excitation levels[8,11]. These high-excitation populations may arise from other non-thermal pumping processes. Burton et al. suggested that $H_2$ formation might be responsible for an apparent excess in the $v = 4$ levels observed in the NGC2023 PDR[16]. In this work, the vibrationally excited $H_2$ formation has been clearly observed in the $H_2O$ photodissociation via the $O(^1S) + H_2$ channel. These $H_2$ fragments have strongly inverted vibrational state population distributions with peaking at $v = 3$ or 4, suggesting an additional source of vibrationally excited $H_2$ in the ISM.

The quantitative determination of the quantum yields of dissociation channels (1)–(6) is needed to assess the importance of both the $H_2O$ photolysis in the VUV region and the role of $H_2(v > 0)$ formation in the interstellar space. This is out of scope in this study. From the references, the quantum yield of the $O(^1D) + H_2$ channel has been measured by Slanger et al.[36] to be 0.1 at around 121.6 nm, whereas McNesby et al.[36,46] reported a yield of 0.14 by detection of $H_2$ from $H_2O$ photolysis. The subtle difference between the two measurements may arise from the contribution of the $O(^1S) + H_2$ channel. Given this assumption, the quantum yield of 0.04 for the $O(^1S) + H_2$ channel can be as the lower limit since this channel should be more and more important as the photoexcitation energy increases. Recent measurements of the H atom PTS[35] and the $O(^1D)/O(^1S)$ fragment ion images are achieved at the photolysis wavelengths of 107.5 nm in our lab. The analysis of these features provides the quantum yield of the $O(^1S) + H_2$ channel to be $\sim16 \pm 8\%$ at 107.5 nm. The large uncertainty comes from the calibration of the detection efficiency of the $O(^1S)$ and $O(^1D)$ atoms (Supplementary Fig. 6 and Supplementary note 3). The average absorption cross-section of $H_2O$ at $\lambda = 100$–112 nm is $\sigma_{H2O} \sim 2 \times 10^{-17}$ $cm^2$[47], enabling estimation of the cross-section for forming $H_2(v > 0)$ fragments at $\lambda = 107.5$ nm: $\sigma \sim 3.2 \times 10^{-18}$ $cm^2$. The abundance of water and VUV photons[48,49] in interstellar space suggests that the contributions of these $H_2(v > 0)$ sources could be significant and thus should be recognized in appropriate interstellar chemistry models.

In summary, the present study provides a benchmark illustration of the use of a FEL-based VUV light source combined with the ion imaging detection methods to exploit the dissociation processes at the state-to-state level, and thereby demonstrates

the exclusive production of vibrationally excited $H_2$ fragments from $H_2O$ photolysis. The single $v$ propensity, accounting for >90% of the total $H_2(X)$ product yield at the photolysis wavelength $\lambda \sim 112.8$ nm, has been observed. This process represents a further source of vibrationally excited $H_2$ observed in the ISM.

## Methods

The vacuum ultraviolet free-electron laser (VUV FEL) has been constructed recently[50]. The VUV FEL facility runs in the high gain harmonic generation mode, in which the seed laser is injected to interact with the electron beam in the modulator. The seeding pulse ($\lambda \sim 240$–360 nm), is generated from a Ti:sapphire laser system. The electron beam is generated from a photocathode RF gun, and accelerated to the beam energy of $\sim$300 MeV by 7 S-band accelerator structures, with a bunch charge of 500 pC. The micro-bunched beam is then sent through the radiator, which is tuned to the $n$th harmonic of the seed wavelength, and coherent FEL radiation with wavelength $\lambda/n$ is emitted. Optimization of the linear accelerator yields a high-quality light beam with emittance of $\sim$1.5 mm·mrad, energy spread of $\sim$1‰, and pulse duration of $\sim$1.5 ps. In this work, the VUV-FEL operates at 10 Hz, and the maximum pulse energy is >100 μJ/pulse. The output wavelength is continuously tunable in the range 50–150 nm.

The VUV FEL-TSVMI experiment involves a molecular beam, photolysis and probe lasers and the detection system, and is summarized in Fig. 1. The pulsed supersonic beam was generated by expanding a mixture of 3% $H_2O$ and Ar at a backing pressure of 1 bar into the source chamber where it was skimmed before entering the ion optics assembly (IOA, 23-plate ion optics[51]) mounted in the differentially pumped reaction chamber. The beam passed through a 2 mm hole in the first electrode and propagates along the center axis of the IOA towards the center of the front face of the detector. The molecular beam was intersected at $90^o$ angles by the counter-propagating photolysis and probe laser beams between the second and the third plates of the IOA. The photolysis photons were provided by the FEL, operating at 10 Hz with $\varepsilon_{pump}$ fixed in the horizontal plane and thus parallel to the front face of the detector. The $O(^1S)$ photoproducts were probed by one-photon excitation at $\lambda = 121.7$ nm via the autoionization transition of $O^*[2s^22p^3(^2P^0)3\,s(^1P^0_1)]\leftarrow O(^1S)$. The 121.7 nm probe photons were generated by four-wave difference frequency mixing (DFWM) using two 212.55 nm photons and one 835.66 nm photon in a cell filled with an Ar/Kr mixture (3:1 mixing ratio). The laser light at 212.55 nm was produced by doubling the output of a 355 nm pumped dye laser operating at $\sim$425 nm. A portion of 532 nm output of the same Nd:YAG laser was used to pump another dye laser which operated at $\sim$835 nm. To eliminate background signals arising from the secondary photolysis of $H_2O$ in the interaction region, the 121.7, 212.55, and 835.66 nm beams were passed through a biconvex LiF lens positioned off-axis at the exit of the Ar/Kr gas cell so as to ensure that only the VUV beam was dispersed through the interaction region.

The resulting $O^+$ ($^1S$) ions were then accelerated through the remainder of the IOA and passed through a 740 mm long field-free region before impacting on a 75 mm-diameter chevron double MCP detector coupled with a P43 phosphor screen. Transient images on the phosphor screen were recorded by a charge-coupled device (CCD) camera (Imager pro plus 2M, La Vision), using a 30 ns gate pulse voltage in order to acquire time-sliced images. Images were taken under different experimental conditions to confirm that the signal was from the intended two-color VUV pump-probe scheme. Specifically, images were recorded: (i) with both pump (VUV FEL) and probe (VUV DFWM) beams present in the interaction region; (ii) with the pump beam present but the probe beam blocked, and (iii) with the pump beam blocked and the probe beam present. The one-color background images recorded under conditions (ii) and (iii), which were very weak compared to the image measured with both beams present, were subtracted from the two-color image recorded under condition (i). Converting the radius of any given feature in the measured images to the corresponding $O(^1S)$ atom velocity relied on calibration factors derived from imaging $O^+$ ions from the one color multiphoton excitation of $O_2$ at $\lambda = 225.00$ nm[52].

## Data availability

The source data underlying Figs. 3 and 4 are provided as a Source Data file. All other data supporting this study are available from the authors upon request. Source data are provided with this paper.

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

## Acknowledgements

The experimental work was supported by the National Natural Science Foundation of China (Grant Nos. 21922306, 21873099), the National Natural Science Foundation of China (NSFC Center for Chemical Dynamics (Grant No. 21688102)), the Key Technology Team of the Chinese Academy of Sciences (Grant No. GJJSTD20190002), the Liaoning Revitalization Talents Program (XLYC1907154). and the international partnership program of Chinese Academy of Sciences (No. 121421KYSB20170012). The theoretical work was supported by the National Natural Science Foundation of China (Grant Nos. 21733006, 22073042, 22122302, and 12047532).

## Author contributions

K.J.Y. designed the experiments. Y.C., Z.J.L., Y.R.Z., and J.Y.Y. performed the experiments. K.J.Y. and Y.C. analyzed the data. K.J.Y., Z.C.C., W.Q.Z., G.R.W., and X.M.Y. discussed the experimental results. F.A., X.X.H., and D.Q.X. performed the theoretical calculation. K.J.Y., Y.C., and X.X.H prepared the manuscript.

## Competing interests

The authors declare no competing interests.

**Additional information**

