## [Peer Review File · Nature Communications]

REVIEWER COMMENTS

Reviewer #1 (Remarks to the Author):

This article describes the discovery of a significant and unexpected channel in the photodissociation of the all important water molecule: $\text{H}_2\text{O} + h\nu$ (102-110 nm) \rightarrow O(1S) + vibrationally excited H₂ molecules. These, as the authors clearly explain, are important and reactive species in astrochemical processes, so discovery of an unknown production method is important. This work is made possible by the unique VUV free electron laser in Dalian, which has enabled an increasing number of discoveries in molecular physics that are published in high profile journals.

This article is very well written (only minor grammar editing is necessary) with clear and necessary figures and complete references. The data analysis is straightforward, especially since the very symmetric O(1S) atom is detected, and the discussion of the dissociation mechanism, while quite basic, is sufficient. Most importantly, the discovery of the O(1S) channel and its H₂(v=3) co-product could be quite important in many communities in science.

On line 193 of the paper the observation of v-propensity is suggested to be unique. However, this channeling to a small subset of vibrational products is discussed in the books of Schinke, and an example of a similar strong propensity can be found in JCP 112 5298 (2000). The mention of this propensity can thus probably be eliminated.

Considering the great expertise of the Dalian group on water photodissociation, a slightly more expanded discussion of why the relative branching ratio for the newly discovered channel is so hard to quantify would be useful. The astrochemical community focuses on these numbers, so a improved strategy on obtaining reliable estimates is important for the future.

These are only minor suggestions and the authors can decide how or not to address them. I recommend publication with optional changes in Nature Chemistry.

Reviewer #2 (Remarks to the Author):

Review of “Vibrationally Excited Hydrogen Molecule Production from the Water Photochemistry: First Observation of the $O(^1S) + H_2$ Channel,” by Yao Chang, Feng An, Zhichao Chen, Zijie Luo, Yarui Zhao, Xixi Hu, Jiayue Yang, Weiqing Zhang, Guorong Wu, Daiqian Xi, Kaijun Yuan, and Xueming Yang

Key Results

There are three major points to the manuscript: 1) the (new) discovery that, in the excitation region from 100-112 nm, water can be dissociated to yield $O(^1S) + H_2(X^1\Sigma_g^-)$, 2) the discovery that the H_2 produced in this reaction is vibrationally excited, and 3) the argument that the cross section for producing vibrationally excited H_2 is high enough so that this mechanism should be included in chemical models of the interstellar medium, where vibrationally excited H_2 has been observed. (Previous models have considered only shock heating and fluorescence decay).

Validity

The experiments are completely convincing in supporting conclusions (1) and (2), above. The arguments concerning conclusion (3) are less well supported.

Significance

If all three conclusions are accepted, the manuscript provides a new explanation for the observed vibrational excitation of H_2 in the interstellar medium and could well account for discrepancies between current physical-chemical models and the observations.

Data and methodology

The data are of high quality. The method of using velocity mapped imaging to measure the results of photodissociation is now almost 35 years old and has been improved here by using time slicing. The apparatus for this part of the experiment is state-of-the-art. The water excitation source, the VUV FEL at Dalian, is key to this experiment. It has also been shown in previous papers (some published in *Nature Communications*) to produce very high quality results. The ionization of the $O(^1S)$ is performed at 121.7 nm using four wave difference frequency generation based on the output of two Nd:YAG-pumped dye lasers. The authors have taken care to minimize unwanted side reactions by dispersing the laser outputs through a biconvex lens so as to remove unwanted frequencies from the interaction region.

Analytical approach

The advantage of the experimental approach is that, once calibrated (as done in this case using O_2 dissociation), the images themselves (Fig. 2) can easily be interpreted to provide solid evidence for the first two key conclusions. The more detailed analysis is based simply on energy conservation equations; it appears to have been done carefully, resulting in Figs. 3 and 4 and Table I. Some details are provided in the Supplementary information (some of the supplementary text explanations (e.g., notes 1 and 2) should be called out in the main manuscript where they are relevant).

Suggested improvements

1. Conclusion (3) is somewhat speculative and either should be stated as such or strengthened by other arguments. To the credit of the authors, they have identified their

assumptions. The most speculative part is the quantum yield for O(¹S) production, whose minimum value is taken as the difference between two very old measurements. The abundance of water and VUV photons should be provided so that the reader can confirm the conclusion (assuming the yield to be correct).

Clarity and context

1. It is possible that a casual reader might misread the title to mean that the O(¹S) produced in water dissociation collides with H₂ to excite it vibrationally. This could be clarified by including the complete reaction in the title: H₂O + hν → O(¹S) + H₂(X¹Σ_g⁺, ν>0).
2. In Supplementary Fig 2 it appears that the population of J=6 is consistently low in ν=2,3, and 4. Any speculation as to what is going on here?
3. Line 205: there is no reason stated for choosing the photoexcitation wavelengths. Presumably these coincide with Rydberg nd states, but this should be clarified (more detail needed).
4. Line 329: the wavelength for ionization here is given as 121.6, whereas it is given as 121.7 earlier in the ms (e.g., line 321).
5. There are many places where the English is ambiguous, wrong, or non-idiomatic. Here are a few:
 - a. Line 73: “has been the long standing problem” should be “has been a long-standing problem”
 - b. Line 94: “some discrepancies yet to be unravelled” should be “some discrepancies are yet to be unravelled”
 - c. Lines 118-119: “band is much complicated” should be “band is much more complicated”
 - d. Line 208: “Similar with the H₂ fragments...” might better be “In similarity with the H₂ fragments...”
 - e. Line 249: “mainly proceeds the transition state”. Do the authors mean “mainly proceeds to the transition state” ?
 - f. Lines 262-263: “like in PDRs where the H₂ (ν>0) molecules are regarded as excitation by FUV fluorescence” Perhaps the authors mean “as in PDRs, where the H₂ (ν>0) molecules are considered to be produced by FUV fluorescence”.
 - g. Line 270: replace “strong” with “strongly”
 - h. Lines 272-274: “ The quantitative assessment ... needs to determine the quantum yields ...” Might this sentence be better as “The quantitative determination of the quantum yields of dissociation channels (1)-(6) is needed to assess the importance of both the H₂O photolysis in the VUV region and the role of H₂(ν>0) formation in interstellar space.”

References

The references seem complete.

Recommendation

Publish after revision.

Response to reviewers' comments

Reviewer #1:

This article describes the discovery of a significant and unexpected channel in the photodissociation of the all important water molecule: $\text{H}_2\text{O} + h\nu$ (102-110 nm) \rightarrow $\text{O}(^1\text{S}) +$ vibrationally excited H_2 molecules. These, as the authors clearly explain, are important and reactive species in astrochemical processes, so discovery of an unknown production method is important. This work is made possible by the unique VUV free electron laser in Dalian, which has enabled an increasing number of discoveries in molecular physics that are published in high profile journals.

This article is very well written (only minor grammar editing is necessary) with clear and necessary figures and complete references. The data analysis is straightforward, especially since the very symmetric $\text{O}(^1\text{S})$ atom is detected, and the discussion of the dissociation mechanism, while quite basic, is sufficient. Most importantly, the discovery of the $\text{O}(^1\text{S})$ channel and its $\text{H}_2(v=3)$ co-product could be quite important in many communities in science.

On line 193 of the paper the observation of v -propensity is suggested to be unique. However, this channeling to a small subset of vibrational products is discussed in the books of Schinke, and an example of a similar strong propensity can be found in JCP 112 5298 (2000). The mention of this propensity can thus probably be eliminated.

Author reply: Thank you for your comments. We have eliminated this description and added the reference (JCP, 112,5298(2000)). Please see page 7, line 9-12.

Considering the great expertise of the Dalian group on water photodissociation, a slightly more expanded discussion of why the relative branching ratio for the newly discovered channel is so hard to quantify would be useful. The astrochemical community focuses on these numbers, so an improved strategy on obtaining reliable estimates is important for the future.

Author reply: Thank you for your comments. We have added an additional discussion about the branching ratio of the observed channel in the main text and also in the SI. Please see page 10 line 12-17 in the main text, and supplementary note 3 in the SI.

These are only minor suggestions and the authors can decide how or not to address them. I recommend publication with optional changes in Nature Chemistry.

Author reply: Thank you again for your comments and for your hard work in our paper reviewing process.

Reviewer #2:

Review of "Vibrationally Excited Hydrogen Molecule Production from the Water Photochemistry: First Observation of the $\text{O}(^1\text{S}) + \text{H}_2$ Channel," by Yao Chang, Feng An, Zhichao Chen, Zijie Luo, Yarui Zhao, Xixi Hu, Jiayue Yang, Weiqing Zhang, Guorong Wu, Daiqian Xi, Kaijun Yuan, and Xueming Yang

Key Results

There are three major points to the manuscript: 1) the (new) discovery that, in the excitation region from 100-112 nm, water can be dissociated to yield $\text{O}(^1\text{S}) + \text{H}_2(\text{X}^1\Sigma_g^+)$, 2) the discovery that the H_2 produced in this reaction is vibrationally excited, and 3) the argument that the cross section for producing vibrationally excited H_2 is high enough so that this mechanism should be included in chemical models of the interstellar medium, where vibrationally excited H_2 has been observed. (Previous models have considered only shock heating and fluorescence decay).

Validity

The experiments are completely convincing in supporting conclusions (1) and (2), above. The arguments concerning conclusion (3) are less well supported.

Significance

If all three conclusions are accepted, the manuscript provides a new explanation for the observed vibrational excitation of H₂ in the interstellar medium and could well account for discrepancies between current physical-chemical models and the observations.

Data and methodology

The data are of high quality. The method of using velocity mapped imaging to measure the results of photodissociation is now almost 35 years old and has been improved here by using time slicing. The apparatus for this part of the experiment is state-of-the-art. The water excitation source, the VUV FEL at Dalian, is key to this experiment. It has also been shown in previous papers (some published in *Nature Communications*) to produce very high quality results. The ionization of the O(¹S) is performed at 121.7 nm using four wave difference frequency generation based on the output of two Nd:YAG-pumped dye lasers. The authors have taken care to minimize unwanted side reactions by dispersing the laser outputs through a biconvex lens so as to remove unwanted frequencies from the interaction region.

Analytical approach

The advantage of the experimental approach is that, once calibrated (as done in this case using O₂ dissociation), the images themselves (Fig. 2) can easily be interpreted to provide solid evidence for the first two key conclusions. The more detailed analysis is based simply on energy conservation equations; it appears to have been done carefully, resulting in Figs. 3 and 4 and Table I. Some details are provided in the Supplementary information (some of the supplementary text explanations (e.g., notes 1 and 2) should be called out in the main manuscript where they are relevant).

Author reply: Thank you for your comments. We have moved some supplementary information from the SI to the main text.

Suggested improvements

1. Conclusion (3) is somewhat speculative and either should be stated as such or strengthened by other arguments. To the credit of the authors, they have identified their assumptions. The most speculative part is the quantum yield for O(¹S) production, whose minimum value is taken as the difference between two very old measurements. The abundance of water and VUV photons should be provided so that the reader can confirm the conclusion (assuming the yield to be correct).

Author reply: Thank you for your comments. We have added the quantum yield estimation in the main text and also in the SI. Recent measurements of the H atom PTS, and the O(¹D) and O(¹S) fragment ion images at the photolysis wavelengths of 107.5 nm in our lab, allow us to estimate the quantum yield of the O(¹S) + H₂ channel at this wavelength. The data analysis provides the quantum yield of the O(¹S) + H₂ channel to be ~16±8% at 107.5 nm. The large uncertainty comes from the calibration of the detection efficiency of the O(¹S) and O(¹D) atoms (Supplementary Fig. 6 and Supplementary note 3).

It is indeed useful to provide the abundance of water and VUV photons, but these values vary at different interstellar circumstances. For instance, recent observations made with the Spitzer Space Telescope show that the water column density of three protoplanetary disks: N(H₂O)≈1.6×10¹⁷ to 8×10¹⁷ cm⁻². While the accumulative luminosity L_{FUV} of the central star is to be 4×10³³ erg s⁻¹ (Science, 326, 1675 (2009)). We have added two references related to the abundance of water and VUV photons in the main text.

Clarity and context

1. It is possible that a casual reader might misread the title to mean that the O(¹S) produced in water dissociation collides with H₂ to excite it vibrationally. This could be clarified by including the complete reaction in the title: H₂O + hν → O(¹S) + H₂(X¹Σ_g⁺, v>0).

Author reply: Thank you for your suggestion, but the formatting requires no punctuation or puns in the title. Thus we have revised the title to “Vibrational excited molecular hydrogen production from the water photochemistry”.

2. In Supplementary Fig 2 it appears that the population of $J=6$ is consistently low in $v=2,3$, and 4. Any speculation as to what is going on here?

Author reply: Thank you for your comments. Due to severe overlapping of the energy levels of the rotational manifolds for $J \geq 6$ and the lower J for the higher vibrational level, the population of $J \geq 6$ is ambiguous. There should be no dynamical reason for the low intensity of $J=6$ in $v=2, 3$, and 4.

3. Line 205: there is no reason stated for choosing the photoexcitation wavelengths. Presumably these coincide with Rydberg nd states, but this should be clarified (more detail needed).

Author reply: Thank you for your comments. We have added some detail to clarify the photoexcitation wavelengths. Please see page 6, line 7-9.

4. Line 329: the wavelength for ionization here is given as 121.6, whereas it is given as 121.7 earlier in the ms (e.g., line 321).

Author reply: Thank you for your comments. We have revised the typos.

5. There are many places where the English is ambiguous, wrong, or non-idiomatic. Here are a few:

a. Line 73: “has been the long standing problem” should be “has been a longstanding problem”

b. Line 94: “some discrepancies yet to be unravelled” should be “some discrepancies are yet to be unravelled”

c. Lines 118-119: “band is much complicated” should be “band is much more complicated”

d. Line 208: “Similar with the H_2 fragments...” might better be “In similarity with the H_2 fragments...”

e. Line 249: “mainly proceeds the transition state”. Do the authors mean “mainly proceeds to the transition state” ?

f. Lines 262-263: “like in PDRs where the H_2 ($v>0$) molecules are regarded as excitation by FUV fluorescence” Perhaps the authors mean “as in PDRs, where the H_2 ($v>0$) molecules are considered to be produced by FUV fluorescence”.

g. Line 270: replace “strong” with “strongly”

h. Lines 272-274: “ The quantitative assessment ... needs to determine the quantum yields ...” Might this sentence be better as “The quantitative determination of the quantum yields of dissociation channels (1)-(6) is needed to assess the importance of both the H_2O photolysis in the VUV region and the role of $H_2(v>0)$ formation in interstellar space.”

Author reply: We thank the reviewer for highlighting these details, all of which we have acted on.

References

The references seem complete.

Recommendation

Publish after revision.

Author reply: Thank you again for your comments and for your hard work in our paper reviewing process.

We very much hope that you judge these responses to be appropriate and hope to see the manuscript published in a future issue of *Nature Communications*.

REVIEWERS' COMMENTS

Reviewer #2 (Remarks to the Author):

The manuscript has been improved by the responses to both reviewers and is now ready for publication.

Response to reviewers' comments

Reviewer #2:

The manuscript has been improved by the responses to both reviewers and is now ready for publication.

Author reply: Thank you very much for your comments.